# MultiLoKo: a multilingual local knowledge benchmark for LLMs spanning 31 languages

**Dieuwke Hupkes**[*]   **Nikolay Bogoychev**[*]
Meta
{dieuwkehupkes,nbogoych}@meta.com

## Abstract

We present MultiLoKo, a benchmark to evaluate multilinguality in LLMs across 31 languages, with three partitions: a `main` partition containing 500 questions per language, separately sourced for each language to be locally relevant, and two `translated` partitions with human-authored translations from 30 non-English languages to English and vice versa. We also release corresponding machine-authored translations. The data is distributed over two splits: `dev`,s and a blind, out-of-distribution `test` split. MultiLoKo can be used to study a variety of questions regarding the multilinguality of LLMs as well as meta-questions about multilingual benchmark creation. We compute scores for 11 base and chat models and study their average performance and performance parity across languages, how much their ability to answer questions depends on the question language, and which languages are most difficult. *None of the models we studied performs well on MultiLoKo*, as indicated by low average scores as well as large differences between the best and worst scoring languages. We also find a *substantial effect of the question language, indicating suboptimal knowledge transfer between languages*. Lastly, we find that using local vs English-translated data can result in differences of *more than 20 points for the best performing models, drastically changing the estimated difficulty of some languages*. For using machine instead of human translations, we find a weaker effect on ordering of language difficulty, a larger difference in model rankings, and a substantial drop in estimated performance for all models.[2]

## 1   Introduction

With the growing presence and deployment of LLMs across the world, evaluating their abilities in languages other than English becomes more and more eminent. Yet, studying and evaluating multilinguality in LLMs is a challenging enterprise, and it is hardly exaggerated to call the current state of multilingual evaluation in LLMs insufficient. Older multilingual benchmarks such as XNLI (Conneau et al., 2018) or XCOPA (Ponti et al., 2020) often do not fit the demands for evaluating auto-regressive models and are rarely used for LLM evaluation. Furthermore, their coverage of languages is relatively small compared to the number of languages in which LLMs are intended to be proficient. More often used are benchmarks translated from English, such as MGSM (Shi et al., 2023) or MMMLU (OpenAI, 2025). These benchmarks provide good coverage over many languages, but using translated data comes with its own set of issues. One such issues is that even when human-rather than machine-authored translations are used, translated data is known to differ from native text in several ways (Clark et al., 2020). Furthermore, using translated benchmarks imposes a strong English-centric bias: translated data may be multilingual on the surface, it is not in its content. The benchmarks MLQA (Lewis et al., 2020) and TidyQA (Clark et al., 2020) to some extent address the

---

[*]Equal contributions
[2]The data, per-language few-shot examples, evaluation scripts, and prompts can be found in our online.

issue by sourcing data separately for different languages. Even in their sourcing protocols, however, there is no explicit focus on selecting locally relevant content for the chosen languages. In addition to that, their coverage is again small compared to the above mentioned translated benchmarks.

In response to these issues, we introduce a wide-coverage multilingual benchmark with locally-sourced questions for 31 different languages. Because the benchmark targets multilingual local knowledge, we dub it MultiLoKo. The release of MultiLoKo serves two interconnected goals:

1) Provide a better means to evaluate multilinguality in LLMs;

2) Provide data to study the effect of various design choices in multilingual evaluation.

To address our first goal, we create 500 questions per language, written from scratch for each language, using a sourcing protocol specifically designed to ensure local relevance of the question topics. To also reap the benefits of parallel data, we commission both human and machine-authored translations for all non-English questions into English and vice versa, providing a total of 15500 parallel questions, sourced across the 31 languages in the benchmark. The translated data facilitates the study of transfer between languages and also serves our second goal. By comparing the English-translated data with the locally sourced data, we can explicitly compare the adequacy of using translated benchmarks; by comparing human- with machine-authored translations, we can better estimate the potential issues of the latter. To prevent quick overfitting and inadvertent contamination, we release a development set of the benchmark, while test scores can only be obtained through an external provider.We compute average performance and language parity scores on the locally sourced data for 11 models marketed for their multilinguality (§ 5.1); we investigate whether models exhibit knowledge transfer between different languages (§ 5.2); we study the impact of local sourcing versus translating on model rankings and language difficulty (§ 5.3.1); we analyse the difficulty of the included languages through various lenses (Appendix D); and we conduct an analysis into the difference between human- and machine-authored translation (Appendix E). We find that the best performing model is Gemini 2.0 Flash, with an average performance of 34.4 points, and an almost 35 point gap between the best and the worst language. Llama 3.1 405B and GPT4-o are close contenders in terms of average scores (34.3 and 34.0, respectively), but both have substantially higher language gaps (39 and 49 points). Almost across the board, model performances are better when questions are asked in the language to which the content is relevant, indicating suboptimal knowledge transfer between languages, a result that is mirrored by low response-consistency across question language.

Next, we study the relevance of using locally sourced data as opposed to translated English data as well as whether it matters if translations are authored by humans or machines. We find that the estimated difficulty of some languages changes drastically across the two sourcing setups, within the range of 15 points decrease and 8 points increase on average across models. The rank correlation between average language difficulty score is 0.78. Furthermore, individual model scores between local and English-translated data can differ up to 22 points for some languages. However, changing the sourcing setup does not impact model rankings, suggesting that using translated data may be suitable for comparing models but less for model development or language prioritisation. For using machine- instead of human-authored translations, as well, the effect on model ranking is limited (R=0.97), but the difficulty estimates of various languages changes with up to 12 points. Furthermore, using machine translated data results in lower average scores for all models, with drops ranging from 2 to 34% of the human-translated scores.

**Outline** In the remainder of this paper, we first describe our dataset collection protocol (§ 2) the dataset itself in (§ 3), and our experimental setup (§ 4). In § 5, we present a range of different results. We conclude in § 6 and discuss limitations in Appendix F. Beyond the related work discussed above, we include a discussion of a wider range of multilingual datasets in Appendix A.

## 2 Dataset collection

Similar to the protocol used by the well-known benchmark SQuAD (Rajpurkar et al., 2016), we source articles from Wikipedia about which we ask annotators to generate questions. After that, we run several rounds of quality control on the generated questions and commission human- and machine-authored translations of all data. Our collection protocol consists of five steps.

**Step 1: Paragraph selection** We start by sampling the 6K most visited Wikipedia pages for each language for the period of 2016-2021. We then sample paragraphs from those pages by randomly selecting a word in the page and expanding left and right until we reach 3K characters. Next, we

ask annotators to judge the local relevance of the samples on a scale from 1 to 5, where 1 refers to topics specific to the language (e.g. a Swedish singer not known outside of Sweden) and 5 to globally well-known topics (e.g. 'Youtube'). We disregard all topics that have a locality score above 3. The full rubric and annotation instructions can be found in Appendix I.1.

**Step 2: Question generation**   In step 2, we ask native speakers to generate challenging questions about the content in the paragraphs. To facilitate automatic scoring, we ask that the questions are closed-form questions, with only one correct short answer. To ensure that the annotation instructions are understandable and appropriate for each locale and the questions of high quality, we run a pilot with 50 questions separately for each language. After our pilot, we commission 500 additional samples for each language, to leave a 10% margin to disregard questions in the rest of the process.

**Step 3: Question review**   For each generated question, we ask a new set of annotators from a separate provider to judge whether the generated questions abide by the annotation instructions, to flag any possible issues, and to mark if the question is useable as is, would be useable with a small adaptation or should be disregarded. We ask annotators to fix small annotation errors on the spot, and as respective vendors that questions with larger issues are replaced.

**Step 4: Question answering**   As a last quality control step, we ask two annotators different from the creator of the question to answer the questions. In this stage, we do not ask annotators to correct questions, but we simply disregard all questions for which either annotator thinks the original answer was incorrect, or the annotator provided an answer not matching the original answer because of ambiguities in the question. The only corrections we allow in this stage are additions of additional, semantically equivalent, correct answers (e.g. 'four' as an alternative to '4').

**Step 5: Translation**   Lastly, we translate the non-English data back to English and vice versa. This allows to study generalisation of knowledge and skills between English and non-English languages and facilitates inspection of the topics and questions for all languages of the dataset, without understanding those languages. We commission both human and machine translations[3] and study their difference as part of our analysis.

## 3   MultiLoKo the dataset

MultiLoKo consists of three main components: i) the collected data; ii) a set of multilingual prompts to prompt base- and chat models; and iii) a set of metrics.

### 3.1   The collected data

The data in MultiLoKo consists of several *partitions* and two *splits*.

**Partitions**   MultiLoKo has one `main` partition, containing locally-soured data for 31 languages, and four `translated` partitions.   Two of the latter are *human*-translated: `human-translated-from-english`, with translations of English data into the 30 other languages in MultiLoKo and `human-translated-to-english`, containing translations of the non-English subsets into English. The other two are *machine*-translated partitions following the same pattern. All partitions contain 500 samples per language – thus in total 15500 samples in the `main` partition, and 15000 samples in all `translated` partitions. Further statistics about the dataset, such as the distribution over answer types and the average prompt length, can be found in in Appendix B.

**Splits**   Each partition is divided equally over two *splits* containing 250 samples per language: a dev split that can be used for development, and a blind `test` split. Until the `test` split is publicly released, results can only be obtained through model submissions.[4] The splits are not random, but constructed such that for each language the most frequently visited pages are in the dev split while the least frequently visited pages are in the `test` split, roughly preserving the distribution of answer types (e.g. number, name, year, etc). The `test` split can thus be seen as an out-of-distribution (ood) split, specifically meant to assess generalisation (which is challenging in the context of LLMs, see e.g. Hupkes et al., 2023). In § 5.3.2, we provide an analysis of the extent to which the split is truly an ood split, by analysing its difficulty. The results reported in the results section of the paper are dev results.

---

[3]For the machine translations, we use the Google Translate sentence based cloud API.

[4]More details can be found on https://github.com/facebookresearch/multiloko/.

## 3.2 Prompts and few-shot examples

In the spirit of getting truly multilingually appropriate results, we design the prompts required to run separately for each language and release them along with the data. The prompts are written by a different linguistic experts for each language, in consultation with the benchmark creators to ensure they are appropriate for LLMs. We provide prompts for base models and chat models that allow for incorporating up to five few-shot examples all of which we provide in our github repository.

## 3.3 Metrics

MultiLoKo has two main metrics and two auxiliary metrics. The two main metrics – *Exact Match accuracy* (EM) and *Gap* – capture the overall performance of MultiLoKo and are computed on the main partition, whereas the two auxiliary metrics – *Mother Tongue Effect* (MTE) and *Locality Effect* (LE) – combine information from different partitions. We provide a cheat-sheet in Table 2.

**EM and Gap**   EM indicates the performance of a model on a single language or averaged across languages, as measured by the percentage of times the post-processed model answer verbatim matches one of the answers in the reference list. Gap, defined as the difference between the best and the worst performing language in the benchmark, is a measure of *parity* across the individual languages within the benchmark. Taken together, EM and Gap provide a good indication of how well a model is faring on MultiLoKo. Because both gap and EM are binary metrics that may be open to false negatives, we also considered the partial match metrics BLEU (Papineni et al., 2002), ChrF (Popović, 2015) and *contains*, but we did not find any different patterns using those metrics.

**MTE**   Because of the 2x2 design of MultiLoKo, in which we translated non-English data back to English and vice versa, we can compute several metrics related to locality of the requested information. MTE is one of such metrics. It expresses the impact of asking a question in a language to which that question is relevant. We quantify MTE (for non-English languages only), as the delta between the EM score of the locally sourced data asked in the corresponding language (e.g. asking a question about a local Bengali radio station in Bengali) and the EM score when the same questions are asked in English. A positive MTE indicates that information is more readily available when it is relevant to the language in which it was asked, whereas a negative MTE indicates that the information is more easily accessible in English. MTE is a measure related to transfer as well as language proficiency.

**LE**   The locality effect (LE) is a measure of how much performance on knowledge tasks is over- or underestimated through the use of using translated English data, as opposed to locally relevant data. We quantify the locality effect as the difference in EM for English translated data and locally sourced data. If for a language the English translated data has as a higher EM, the LE is *positive*, indicating that using English translated data likely *overestimating* a model's ability on providing knowledge for that language. If the LE is *negative* the English translated data may provide an *underestimation* of the score for that language. Note that because we often observe both positive and negative LEs for the 30 non-English languages in MultiLoKo, the average LE across languages may be small, even if the differences for individual languages may be large.

# 4   Experimental setup

We test and showcase our benchmark by running experiments with 11 different models of varying sizes, that were all marketed to have multilingual abilities.

## 4.1 Models

To test the extent to which MultiLoKo provides useful signal across training stages, we consider both base and chat models. The base models we include in our experiments are Llama 3.1 70B and 405B (Dubey et al., 2024), Mixtral 8x22B (team, 2024), and Qwen 2.5 72B (Qwen et al., 2025), the seven chat models are Gemini 2.0 Flash (Google DeepMind, 2024), GPT4-o (OpenAI et al., 2024), Claude 3.5 Sonnet (Anthropic, 2025), Llama 3.1 70B and 405B Chat, Mixtral 8x22B-it, and Qwen 2.5 72B instruct. As mentioned before, we run chat and base models with separate prompts.

Table 1: **Aggregate results dev.** We report average EM, gap, mother tongue effect and locality effect for all 11 models on the MultiLoKo dev split. For EM, MTE and LE, we also indicate a confidence interval equal to two times the standard error across languages. Models are sorted by average EM.

| Model | EM | Gap | Mother tongue effect | Locality effect |
|---|---|---|---|---|
| Gemini 2.0 Flash | 34.39± 2.90 | 34.80 | 6.12± 1.90 | 0.36± 3.40 |
| Llama 3.1 405B | 34.31± 2.70 | 39.20 | 6.37± 1.70 | 0.62± 2.70 |
| GPT4-o | 33.97± 3.60 | 48.80 | 3.08± 2.00 | 0.35± 2.90 |
| Llama 3.1 405B Chat | 27.70± 3.20 | 40.80 | 3.97± 2.20 | −1.11± 2.70 |
| Llama 3.1 70B | 26.92± 2.60 | 28.80 | 2.72± 1.70 | −0.30± 3.10 |
| Claude 3.5 Sonnet | 26.89± 4.40 | 47.60 | N/A | 0.81± 2.90 |
| Llama 3.1 70B Chat | 21.65± 2.80 | 42.40 | 0.49± 1.60 | −3.32± 3.30 |
| Mixtral 8x22B | 21.64± 4.20 | 43.60 | −2.18± 3.00 | −0.65± 2.60 |
| Qwen2.5 72B | 19.66± 2.30 | 28.40 | 2.45± 2.10 | −2.28± 2.70 |
| Mixtral 8x22B-it | 10.10± 3.10 | 39.20 | −5.41± 2.00 | −0.54± 1.70 |
| Qwen2.5 72B instruct | 2.54± 0.70 | 8.00 | −1.52± 1.00 | 0.43± 0.70 |

## 4.2 Experimental setup

We run all experiments with generation temperature set to 0. To facilitate automatic evaluation, we include an instruction to answer questions curtly and precisely, producing only a number/name/etc. Full template information can be found in our github repository. We use a 5-shot prompt for base models and a 0-shot prompt for chat-models. For base models, minimal postprocessing is needed: we lowercase the output, strip punctuation and whitespace, and evaluate the first line. Chat models often deviate from the required format, in various ways that we discuss in Appendix G. To evaluate such models beyond their instruction-following issues, we perform more complex post-processing, aiming to remove any words resembling "answer" from the LLM output, as well as several special cases for English and Japanese. We provide full details about post-processing in Appendix H.

## 5 Results

We report average model results (§ 5.1), study transfer between languages (§ 5.2) and look in more detail at the dataset itself through the lens of model results (§ 5.3). We report language specific results and differences between using human and machine translated data in Appendices D and E.

## 5.1 Aggregate results: EM and language gap

In Table 1, we report per-model average EM, the gap between the best and worst language, and average MTE and LE, which we will discuss in a later section. We report average MTE, EM and LE along with a confidence interval equal to two times the standard error across languages, roughly equalling previously used 95% confidence intervals (Madaan et al., 2024; Dubey et al., 2024).

**Model performance (EM)** In Figure 1 (left), we show a boxplot of the EM scores across models. The best performing models are Gemini 2.0 Flash, Llama 3.1 405B, and GPT4-o, while Mixtral 8x22B and Qwen2.5 72B populate the lower rankings. Somewhat surprisingly, base models are generally outperforming chat models on the benchmark, this is partly due to false refusals and poor instruction following in the chat models. In some cases, however, the chat models simply just provide a qualitatively different answer than the base models. The figure shows that MultiLoKo is a relatively difficult benchmark across the board: the average EM of even the best performing model barely exceeds 30, while the bottom performing models have scores lower than 20. Also EM for the easiest languages (see also Appendix D) remain below 50. Furthermore, for virtually all models performance varies starkly between languages, suggesting that none of the models we considered are evenly multilingual across the 31 languages covered.

**Gap** While average EM score provides some information about a model's multilingual abilities, the same EM score can hide many different patterns regarding individual language scores. As we appreciate it is not always practical to consider 31 separate EM scores in model development, we add a second summary metric to the main metrics of MultiLoKo: the gap between the best and worst performing languages, reperesentative of the extent to which a model has achieved parity across languages. Earlier, we already saw that the per-language scores have quite a range for all models. In

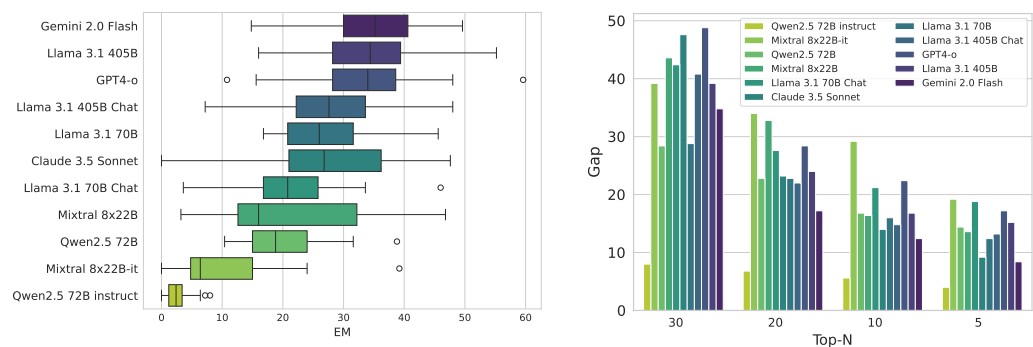

Figure 1: **EM distributions and Gap dev.** Left: Boxplot of EM scores across models, sorted by mean. Right: Difference between the best EM and the worst of the N next best EM scores, per model.

Figure 1 (right), we study this in more detail, by considering the gap between the best language and the next N best language (30 corresponds to the full benchmark). On the right end of the plot, we see that already considering only 5 languages besides English, even the best perform has a gap of over five points – which is relatively large in absolute terms and very large in relative ones – between English and the worst of the remaining languages. For the second best two models, the top-5 gap even exceeds 10 points. As we include more languages, up to the full benchmark, the gap increases, with GPT4-0 showing gap of almost 50 points. The only models for which the gap is small are the models that have overall low performance and thus little space to drop from English, illustrating how gap and average EM provide complementary information about multilingual performance.

## 5.2 Generalisation across languages

Next, we study whether knowledge generalises across languages.

**The mother tongue effect (MTE)** First, we compare the EM of models when questions are asked in the language for which the questions were originally sourced with performance when the same questions are asked in English. We quantify this effect with the metric MTE, which expresses the difference in performance between these two settings (see § 3.3). In Figure 2 (left), we show MTE per language, averaged across models.[5] For most languages, performance is higher when the question is asked in the language for which the question is locally relevant. With the exception of Hindi, the languages for which MTE is negative or close to 0 are all languages that perform very poorly also in the mother tongue and for which there is therefore little room for further decrease. From one perspective, the improvements when questions are asked in the low-resource but native languages can be seen as surprising: as models perform much better in English than non-English languages, one may expect performances to go up as a consequence of that. On the other hand, similar 'mother tongue effects' have been observed in earlier studies. For example, Ohmer et al. (2024) found that models are comparatively better at answering factual questions about topics when they are asked in a language to which culture the fact pertains. It appears that also in our case, the effect of accessibility of information in a relevant language wins out over the generally stronger English performance, pointing to a gap in models' ability to generalise knowledge from one language to another. In Figure 2 (right), we further consider the distribution of MTE scores for the top-3 models. Interestingly, this distribution is quite different between models. Despite having comparable average scores, the top-3 performing models differ in their MTE distributions across languages. Of the three models, GPT4-o has the smallest average effect (3.2); Llama 3.1 405B has a much higher average effect (6.6), but less probability mass on the more extreme ranges of the spectrum (min max values of [-7, +12] vs [-9, +13]) Gemini 2.0 Flash is in the middle in terms of average (6.3), but shows the largest variation across languages ([-10, +16]). Note, however, that without studying the actual training data of the models, it is possible to infer that the models have relatively poor transfer across languages, but not conclusively say that one model is better than another: it is also possible that the information sourced for languages with better MTEs was simply better represented in the English data of a model.

---

[5]Claude 3.5 Sonnet scores were very low on English because of poor instruction following (s see Appendix G). As this is unrelated to lack of transfer or knowledge, we exclude Claude 3.5 Sonnet from all transfer results.

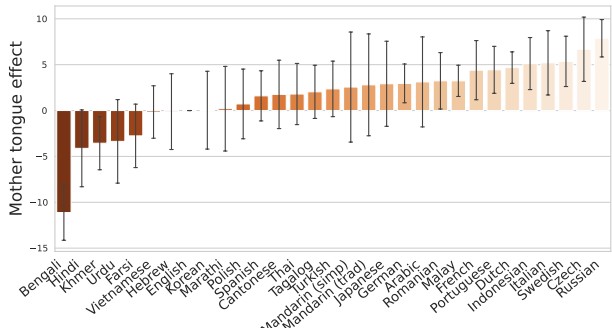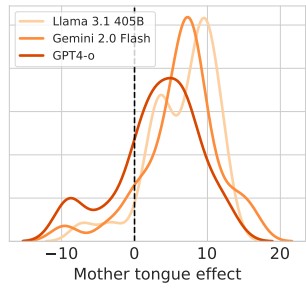

Figure 2: **Mother tongue effect.** Left: Per language MTE, indicating the delta between asking questions in mother tongue vs English. Error bars indicate 2x SE across all models but Claude 3.5 Sonnet. Right: KDE plot of the distribution of MTE scores for the top-3 performing models.

| Model | Consistency |
|---|---|
| Gemini 2.0 Flash | 0.46± 0.04 |
| Llama 3.1 405B | 0.46± 0.04 |
| Llama 3.1 70B | 0.45± 0.03 |
| GPT4-o | 0.45± 0.05 |
| Llama 3.1 405B Chat | 0.42± 0.04 |
| Qwen2.5 72B | 0.40± 0.04 |
| Llama 3.1 70B Chat | 0.40± 0.04 |
| Mixtral 8x22B | 0.36± 0.05 |
| Mixtral 8x22B-it | 0.21± 0.05 |
| Qwen2.5 72B instruct | 0.08± 0.03 |

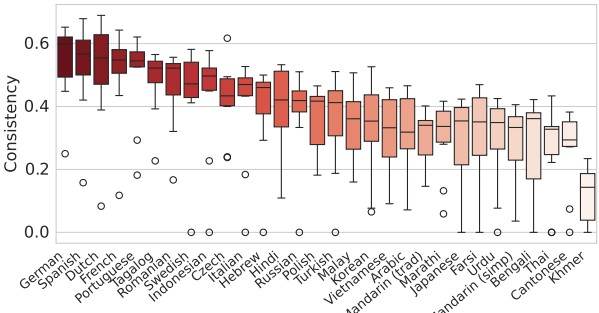

Figure 3: **Consistency results.** Left: Average per-model consistency scores, ± 2 times the standard error across languages. Right: Boxplot of model consistency scores per language, indicating the relative overlap of correctly answered questions when asked in the mother tongue vs in English.

**Consistency across responses**    Another way to study transfer between languages is to look at the consistency of responses across languages (Qi et al., 2023; Ohmer et al., 2023, i.a.). After all, it is possible for a model with an EM of 30 on both native and translated data to be completely misaligned on *which* questions they respond to correctly. Studying consistency across responses can therefore be seen as a more direct way of studying whether knowledge is equally accessible across languages. In the dataset used by Ohmer et al. (2023), the correct answers are identical across languages, while Qi et al. (2023) use a ranking approach. Neither of their metrics can be directly applied in our case. Rather, we opt for a simpler consistency metric, which quantifies what percentage of the questions that are answered correctly in *either* language are answered correctly in *both* languages. In Figure 3 (left), we show the average consistency of all models; we also show the per-language consistency results in Figure 3 (right). The results confirm our earlier conclusion that much improvements can be made when it comes to knowledge transfer between languages: even for the best performing models, there is an overlap of not even 50% between the questions correctly answered across languages.

## 5.3    The dataset

Lastly, we discuss two aspects related to the creation of the dataset.

### 5.3.1    Locally-sourced vs translated-from-English data

To study the impact of using locally sourced data, we consider the difference between per-language EM on locally sourced data and translated from English data.

**Language difficulty**    First, we look at per-language differences between locally sourced and translated English data. We quantify this with a metric we call the Locality Effect (LE), which tells us how much the estimate of a model's strength in a particular language would have been off if we had chosen to use a translated benchmark rather than a locally sourced one. We plot this difference

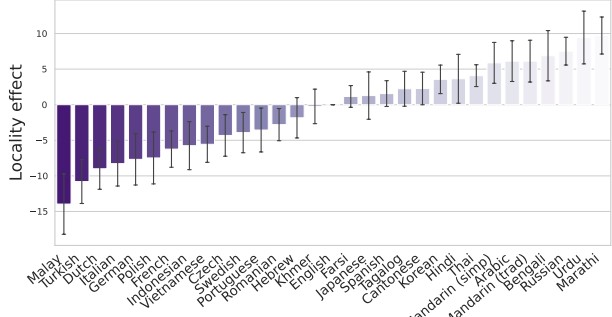

| Model | Rank correlation language difficulty |
|---|---|
| Gemini 2.0 Flash | 0.54 |
| Llama 3.1 405B | 0.65 |
| GPT4-o | 0.64 |
| Llama 3.1 405B Chat | 0.70 |
| Llama 3.1 70B | 0.60 |
| Claude 3.5 Sonnet | 0.84 |
| Llama 3.1 70B Chat | 0.68 |
| Mixtral 8x22B | 0.86 |
| Qwen2.5 72B | 0.45 |
| Mixtral 8x22B-it | 0.88 |
| Qwen2.5 72B instruct | 0.55 |

Figure 4: **Locality Effect.** Left: Per language LE, expressing the delta in EM between locally sourced and translated English data. Right: Per-model rank correlation between language difficulty of languages on locally sourced vs English translated data.

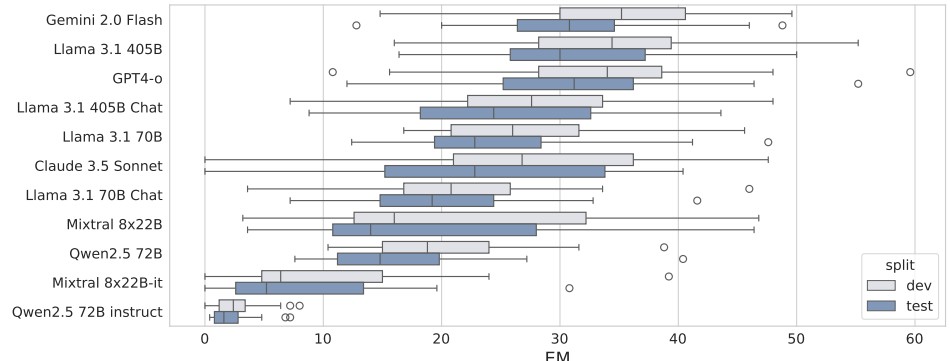

Figure 5: **Average EM, dev versus `test`.** Score distributions of the dev (upper bars) and `test` (lower bars) sets. The results show that the `test` set is indeed out of distribution with respect to the dev set.

in § 5.3.1 (left). As we can see, the scores between locally and translated English-sourced data can differ quite drastically, almost 15 percentage points averaged across models. For individual models, the differences are even larger. For Llama 3.1 405B, LE ranges from -13 to +17; for Gemini 2.0 Flash from -21 to +15; and for GPT4-o from -22 to +14. The differences are not just in absolute scores; also the ordering of language by difficulty is quite different across the two data collection setups, as can be seen by the per-model rank correlations of language difficulty between the two conditions, shown in § 5.3.1 (right). Using English-translated rather than locally sourced data does thus not only provide different estimates, but may suggest different languages to focus on for improvement.

**Model rankings** Next, we consider the ranking of the models under the two different data regimes. Interestingly, given the transfer effect, changing from locally to English translated data does not make any difference in the ranking. Also in terms of absolute scores, the difference between the two data collection setups is relatively minor. At least for our type of data, it thus appears that using translated data as opposed to locally sourced data may be a reasonable setup for comparing models on average, though not for getting adequate per-language or set language prioritisation.

### 5.3.2 The dataset split

As mentioned in the dataset construction, we took the deliberate decision to generate a split based on topic frequency, rather than creating a random split. The aim of this out-of-distribution split is to test generalisation to topics that are more in the tail of the distribution, as well as encourage improvements in multilinguality beyond having a higher score on the specific released MultiLoKo dev set. Of course, however, because of our sourcing method, *all* the topics in MultiLoKo are topics on which information is available on Wikipedia. As training data, Wikipedia is often packaged as a single scrape, this may render our deliberate splitting efforts futile: the fact that a page is less visited does not make it less likely that the specific page is included in the training data. Now, we test if the dev and `test` split are in fact distributionally different.

In Figure 5, we show boxplots of dev and `test` EM scores for all models under consideration. The plot confirms that the split is indeed to be considered an OOD split: for virtually much all models, the test scores are lower than the dev scores. Across all models, the average dev score is 24, whereas the average test score is 21. This suggests that our test set does indeed contain more tail knowledge than the dev set, despite the aforementioned arguments regarding Wikipedia. Interestingly, this implies that Wikipedia may not be the primary source from which models learn this information.

The difference in difficulty also has bearing on the other metrics: the *gap* between the best and worst performing language) is 37 for dev vs 34 for `test`, suggesting that more difficult dat may to some extent hide differences between languages and therefore exemplifying the utility of considering parity along with overall performance. The mother tongue effect, on the other hand, is comparable across dev and `test` (1.61 vs 1.56, respectively). For the locality effect, the effect is less interpretable. While the average difference is substantial (-0.6 dev vs -1.9 `test`), there is no clear pattern discernable across languages: for some, the effect reduces, whereas for others it increases.

# 6   Conclusion

Notwithstanding the increasing multinational deployment of LLMs in many parts of the world, adequately evaluating their multilinguality remains a challenging enterprise. Only in part is this due to the scarcity of high-quality and broad-coverage multilingual benchmarks for LLM: perhaps a more pressing issue is that the benchmarks that *are* frequently used for multilingual evaluation virtually all consist of translated English data. While using completely parallel data has its advantages, using translated English data imposes an English-centric bias on the content of the benchmarks, implying that even if the benchmark evaluates multilinguality on the surface, it does not in content. In our work, we aim to address this by presenting `MultiLoKo`, a multilingual benchmark spanning 31 languages that combines the best of both worlds. MultiLoKo contains 500 questions targeting locally relevant knowledge for 31 languages, separately sourced for each language with a protocol specifically designed to ensure local relevance of the question topics. It is also fully parallel, because it contains human-authored translations of the non-English partitions into English and vice versa. As such, it allows to study various questions related to multilinguality, transfer and multilingual benchmark creation. To prevent quick overfitting and inadvertent contamination, we release a development set of the benchmark, while the test set of the benchmarks remains private, at least for the near future.

We use MultiLoKo to analyse 4 base and 7 chat models marketed to be multilingual. We find that the best performing model is Gemini 2.0 Flash, with an average performance of 34.4 points, and an almost 35 point gap between the best and the worst language, followed by Llama 3.1 405B and GPT4-o, which are close contenders in terms of average performance but both have substantially higher language gaps (39 and 49 points). Generally, scores are better when questions are asked in the language to which they are relevant, indicating suboptimal knowledge transfer between languages, a result that is mirrored by low per-sample consistency across question language.

On a meta-level, we study the relevance of using locally sourced data as opposed to translated English data as well as whether it matters if translations are machine- or human-authored. We find that the estimated difficulty of some languages changes drastically across the two sourcing setups, within the range of 15 points decrease and 8 points increase on average. The rank correlation between average language difficulty score is 0.78, and individual model scores can differ up to 22 points for some languages. However, changing the sourcing setup does not impact model rankings, suggesting that using translated data may be suitable for comparing models but less for model development or language prioritisation. For using machine- instead of human-authored translations, as well, the effect on model ranking is limited (R=0.97), but the difficulty estimates of various languages changes with up to 12 points. Furthermore, using machine translated data results in lower average scores for all models, with drops ranging from 2 to 34% of the human-translated scores.

While our results section is extensive already, there are still several parts of MultiLoKo that we did not explore. For instance, because of the sourcing strategy, each native question is coupled with a paragraph that contains the answer to the question. MultiLoKo could thus be transformed into a reading-comprehension benchmark, and we consider studying the difference between the knowledge and reading comprehension setup an interesting direction for future work. Furthermore, each question contains an elaborate long answer intended to explain the short answer. We have not used the long answers in any of our experiments, but foresee interesting directions including studies into CoT prompting or studying answer rationales.

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

| | |
|---|---|
| *Average EM* | The first main metric we use to quantify performance for MultiLoKo is the average Exact Match score across languages, which expresses how many of the answers match one of the gold standard answers verbatim (after post-processing the answers). |
| *Gap* | The second main metric is the gap between a model's best and worst performing language. We gap to quantify the extent to which a model has achieved *parity* across languages. Because a small gap can be achieved both through parity on high scores as parity on low scores, it is most informative in combination with average benchmark performance. |
| *Mother tongue effect (MTE)* | MTE expresses the impact of asking questions in a language in which the requested information is locally salient, compared to asking it in English. A positive MTE indicates information is more readily available in the language it was (likely) present in the training data, whereas a negative mother tongue effect indicates the information is more easily accessible in English. |
| *Locality effect (LE)* | LE quantifies the effect of using locally sourced vs translated data. It is measured by computing the difference between scores for locally sourced data and translated English-sourced data. A positive LE implies that using translated English data *underestimates* performance on a language, a negative LE that using translated English data *overestimates* performance. |

Table 2: **MultiLoKo metric cheatsheet.** We use several metrics to quantify model performance using MultiLoKo. This table provides a cheatsheet for their meaning.

## A Related work

In this paper, we introduce a new multilingual benchmark for LLMs, that we believe addresses gaps and pitfalls in existing benchmarks. We (concisely) outlined those gaps and pitfalls and mentioned several other works related to ours in the introduction of those paper. Here, we discuss multilingual evaluation of LLMs in more detail. Specifically, we discuss what datasets recent LLM releases have used for multilingual evaluation (Appendix A.1) and what other datasets and approaches they could have used but did not (Appendix A.2).

### A.1 Multilingual evaluation of LLMs in practice

While multilinguality is something frequently mentioned in the release papers or posts of recent LLM releases, the datasets for which they actual report scores is in most cases quite limited. Of the models that we evaluated for this paper, Gemini 2.0 Flash reported no multilingual scores at all; GPT4-o and Mixtral 8x22B report scores only on internally translated but not publicly available English benchmarks; Claude 3.5 Sonnet reports scores for only one benchmark – MGSM. MGSM is also the only publicly available benchmark for which Llama 3.1 reports scores, along with – also – an internally translated version of MMLU that is not publicly available. The only model that extensively reports multilingual benchmark values, on more than 10 benchmarks, is Qwen2.5 72B. We provide an overview of the multilingual benchmarks for which scores are reported for these models in Table 3.

| | |
|---|---|
| Claude 3.5 Sonnet | MGSM (Shi et al., 2023) |
| Gemini 2.0 Flash | Mentions multilingual audio, no multilingual benchmarks scores reported. |
| GPT4-o | ARC-Easy and TruthfulQA translated into five African languages (internal benchmark), Uhura-Eval (internal benchmark). |
| Llama 3.1 | MGSM (Shi et al., 2023), Multilingual MMLU (internal benchmark) |
| Mixtral 8x22B | translated ARC-C, HellaSwag and MMLU (internal benchmarks) |
| Qwen2.5 72B | M3Exam (Zhang et al., 2023), IndoMMLU (Koto et al., 2023), ruMMLU (Fenogenova et al., 2024), translated MMLU (Chen et al., 2023), Belebele (Bandarkar et al., 2024), XCOPA (Ponti et al., 2020), XWinograd (Muennighoff et al., 2023), XStoryClose (Lin et al., 2022), PAWS-X (Zhang et al., 2019), MGSM (Shi et al., 2023), Flores-101 (Goyal et al., 2022) |

Table 3: **Multilingual evaluation of recent LLM releases, overview.** We provide an overview table of the benchmark for which scores are reported in the release papers or notes of the LLMs we evaluated in this paper. Models are sorted alphabetically.

## A.2 Multilingual evaluation options for LLMs

While, as we discuss below, there are gaps and challenges with multilingual evaluation for LLMs, there are in fact many more options than is suggested by what is reported in recent releases. Below, we discuss other options for multilingual LLM evaluation.

**Translated English benchmarks** As mentioned earlier on, benchmarks used for LLM evaluation are often translated English benchmarks. In some cases, the benchmarks were designed to evaluate only English and translated later, such as translated MMLU (e.g. Li et al., 2024; Chen et al., 2023; OpenAI, 2025; Singh et al., 2024) or MMLU-ProX (Xuan et al., 2025), MGSM (Shi et al., 2023) or MLAMA (Kassner et al., 2021). In other cases, the benchmark was multilingual at the time of its creation, but means of creation of the non-English data was through translating English sourced data, such as Belebele Bandarkar et al. (2024), Mintaka (Sen et al., 2022), or X-FACTR (Jiang et al., 2020). Taken together, translated benchmarks span quite a range of tasks, such as question answering (Artetxe et al., 2020; Lewis et al., 2020; Qi et al., 2023; Ohmer et al., 2023), natural language inference (Conneau et al., 2018), paraphrase detection (Zhang et al., 2019), general linguistic competence (Jumelet et al., 2025), reading comprehension (Artetxe et al., 2020; Bandarkar et al., 2024) and commonsense reasoning (Ponti et al., 2020), and even instruction following (He et al., 2024). With the exception of question answering and of course instruction following, however, many of these tasks have gone (somewhat) out of fashion for LLM evaluation, a trend which is mirrored also in the usage of their multilingual counterparts. As mentioned before, translated benchmarks have the advantage of containing parallel data, allowing for some form of comparability across languages, but are English-centric in content and may suffer from translationese (see e.g. Romanou et al., 2024; Chen et al., 2024, for a recent discussion of this).

**Multilingual benchmarks sourced from scratch** Though much rarer, there are also benchmarks that are created independently for each language they include. Clark et al. (2020) release a question answering dataset separately sourced for 11 different languages, with a protocol relatively similar to ours. In a different category, Hardalov et al. (2020), Zhang et al. (2023) and Romanou et al. (2024) and Sánchez et al. (2024) do not *create* benchmark data, but instead collect existing exam or competition questions from official human exams. In case of Zhang et al. (2023), the exams are graduation exams of primary, middle and high school; Hardalov et al. (2020) includes official state exams taken by graduating high school students, which may contain parallel pairs in case countries allow examinations to be taken in multiple languages; Romanou et al. (2024), cover academic exams at middle and high school and university level, professional certifications and licenses, and exams to obtain regional licenses. Sánchez et al. (2024) instead focus on questions from the International Linguistic Olympiad corpus. Lastly, as part of their study Ohmer et al. (2023) create a dataset called SIMPLE FACTS, containing factual questions created through a shared template filled in with language specific factual data.

**Consistency evaluation** A rather different approach to assess multilinguality in LLMs is to focus not on accuracy across different languages, but to consider whether predictions are *consistent* across languages. This tests knowledge and skill transfer between languages more explicitly. Two recent examples of studies incorporating consistency-based evaluations on factual knowledge questions are Qi et al. (2023) and Ohmer et al. (2023). Qi et al. (2023) focusses specifically on sample-level consistency of answers across different languages, requiring existing parallel benchmarks. Ohmer et al. (2023), instead, ask models to translate benchmark questions themselves before answering them again. This can, with some caveats, be applied to any existing monolingual benchmark, but – requiring multiple steps – it is more involved an a paradigm, and is somewhat bottlenecked by the translation ability of the model to be evaluated.

**Translation as a proxy for multilinguality** Another, more implicit method to assess multilinguality in LLMs is to evaluate their ability to translate from one language to another. This approach was famously used by Brown et al. (2020), but has not been common since.

**Monolingual non-English evaluation** In our discussion, we have focussed on multilingual evaluation options that cover multiple other languages. After all, a benchmark to evaluate models on Bengali (e.g. Shafayat et al., 2024) or Arabic (e.g. Alwajih et al., 2024) can contribute to multilingual evaluation when combined with other benchmarks, but does not so on its own. Because such

benchmarks are usually created by language experts for the respective languages, they usually target locally relevant skills and knowledge and are likely of higher quality than benchmarks created for many languages simultaneously (either through translation or from scratch). Yet, composing a suite including many languages that allows direct comparisons between languages remains challenging. We believe such benchmarks can be important for multilingual evaluation in LLMs, but will not further discuss benchmarks focussing on individual languages or very small sets of languages within one family here.

# B  Additional dataset statistics

For reference, we provide a few dataset statistics beyond the main results in the paper.

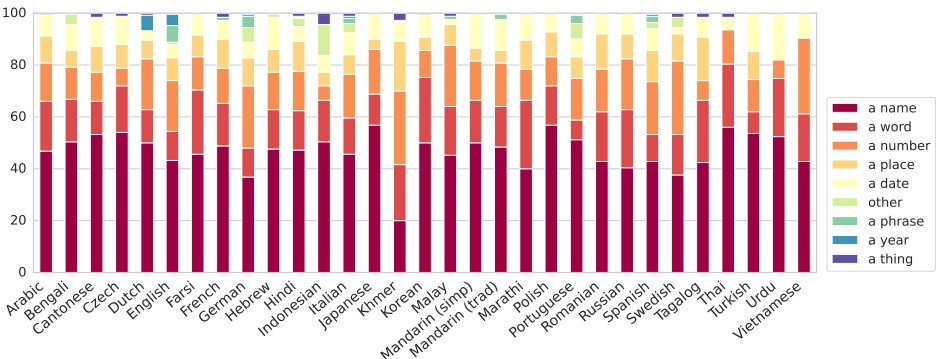

Figure 6: **Distribution of output types on the dev split.** We show the normalised distribution of correct output types across languages, ordered (from bottom to top) by average frequency. Rare output types that occur only a few times are mapped to the category 'other'.

**Output type distribution**    In Figure 6, we show the per-language distribution of output types for MultiLoKo dev split.[6] We mapped very rare output types, such as 'a quantity', 'a period of time' or 'letter' to 'other', for plotting purposes. We can see that *name* is the most common output type across languages, followed by the generic output type *a word* and *number*. Also *place* and *date* are relatively common output types, whereas most other output types occur very infrequently or only for a handful of languages.

**Input and output length**    In addition to that, we show the average question – and output lengths of human-translated the locally sourced questions to English in Figure 7. While there is some variation in particular in question length, the lengths of the answers are relatively consistent. The average answer length is around 2, combining one-word answers with (usually) longer names.

# C  Model runtime and compute resource information

In this section we provide details about the runtime for non-API models (LLaMa 3 family of models, Qwen and Mixtral).

| | |
|---|---|
| Hardware | All models were run on one node of 8xH100 80GBs, 1 TB of RAM, with the exception of LLama 3.1 405B, for which we used 2 nodes. |
| Precision | All models were run on **bf16** precision. |
| Runtime | Excluding model loading time, all models took <10 minutes to complete a partition of the dataset. A partition is defined as 250 examples, and could be either dev, test, or human/machine translated version of those. Overall, < 1 hour is required to run every single example in our dataset through an LLM with this setup. |

For iterating over prompts and debugging, we used a reduced dataset of 50 examples, and a 70B model, which overall took less than 1 hour of total compute resources.

---

[6]Because the test split is blind, we do not report the distribution of output types here.

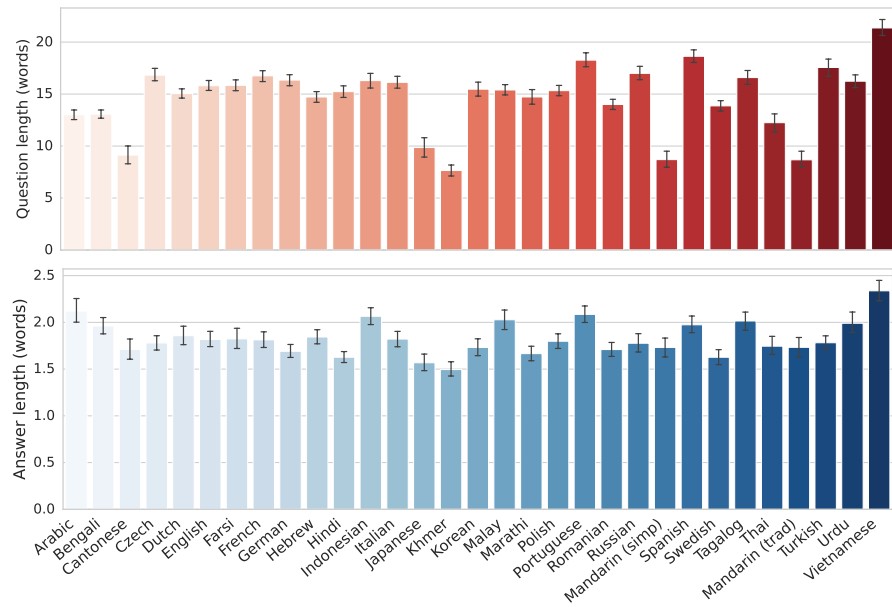

Figure 7: **Average question and answer lengths.** We show the per-question average length (in words) of the locally-sourced questions and answers, human-translated into English.

## D  Differences between languages

So far, with the exception of MTE and parity scores, we have primarily looked at results averaged across languages. Now, we consider language-specific results in a bit more detail.

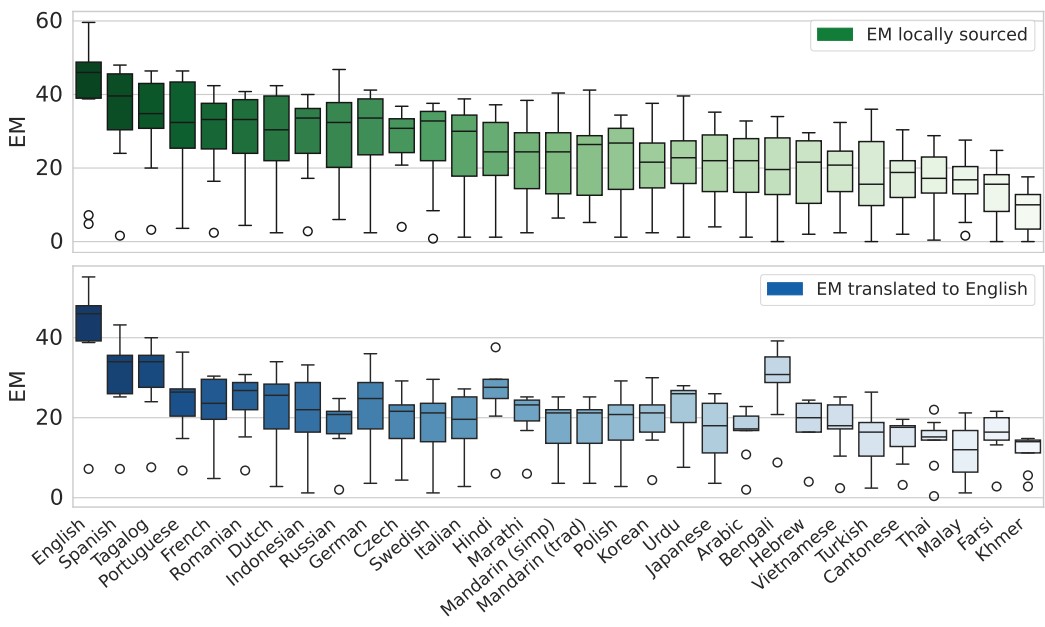

Figure 8: **Average EM per language dev, in mother tongue and English.** Top: Average EM on locally sourced data. Bottom: Average EM on locally sourced data, translated to English.

### D.1 Language difficulty on locally sourced data

First, in Figure 8 (top), we show average results for all languages on all locally sourced data. In broad strokes, the order of difficulty is correlated with how low- or high- resource a language is considered: while languages such as French, English and Spanish occur at the easier end of the spectrum, we find Farsi, Khmer and Malay among the most difficult languages. There are a few notable exceptions: on average the second highest scoring language in our benchmark is Tagalog. While it is difficult to judge why without doing a detailed analysis on the questions, we hypothesise that the questions asked by the Tagalog language experts are simply less complex than the questions of other languages.

### D.2 Separating language difficulty from language proficiency

In an attempt to distinguish *data difficulty* from *language proficiency*, we consider also the difficulty of the locally sourced data translated to English. While this conflates data difficulty and transfer (see § 5.2), it still gives us some indication of the extent to which low performance in languages is caused by poor language proficiency versus data difficulty. In the bottom half of Figure 8, we show the model performances as computed on the locally sourced data *translated to English*. The correlation between these two language difficulty rankings between these setups is 0.79. When comparing the ranks of the various languages, only a handful of languages shift more than a few places. Specifically, Bengali (26->4), Urdu (26->12), and Hindi (14->5) all decrease substantially in difficulty rank. The fact that they are comparatively easier in English suggests that for those languages proficiency may be a larger problem than data difficulty. On the other hand, only Russian (7->21) shows a drop of more than 5 places.

## E Human versus machine translation

In this section, we consider the impact of using machine- or human-authored translations. To do so, we look at the differences in EM scores between machine and human translated data for the various languages, taking the human translations as the 'gold standard' (i.e. we consider human translated EM - machine translated EM). We show the results in Figure 9.

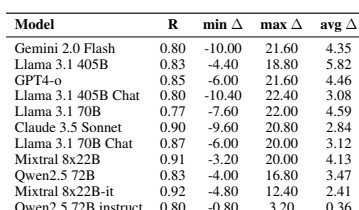

| Model | R | min Δ | max Δ | avg Δ |
|---|---|---|---|---|
| Gemini 2.0 Flash | 0.80 | -10.00 | 21.60 | 4.35 |
| Llama 3.1 405B | 0.83 | -4.40 | 18.80 | 5.82 |
| GPT4-o | 0.85 | -6.00 | 21.60 | 4.46 |
| Llama 3.1 405B Chat | 0.80 | -10.40 | 22.40 | 3.08 |
| Llama 3.1 70B | 0.77 | -7.60 | 22.00 | 4.59 |
| Claude 3.5 Sonnet | 0.90 | -9.60 | 20.80 | 2.84 |
| Llama 3.1 70B Chat | 0.87 | -6.00 | 20.00 | 3.12 |
| Mixtral 8x22B | 0.91 | -3.20 | 20.00 | 4.13 |
| Qwen2.5 72B | 0.83 | -4.00 | 16.80 | 3.47 |
| Mixtral 8x22B-it | 0.92 | -4.80 | 12.40 | 2.41 |
| Qwen2.5 72B instruct | 0.80 | -0.80 | 3.20 | 0.36 |

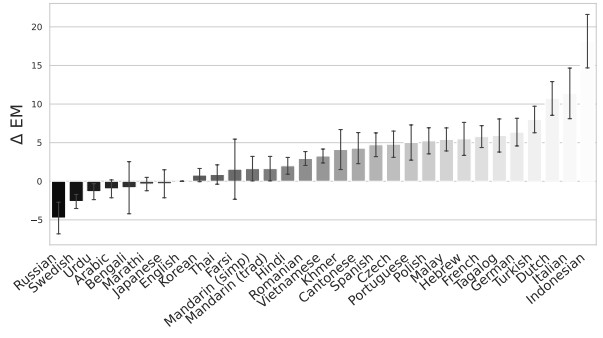

(a) Language difficulty stats
across human- and machine translations

(b) MT vs human translations

Figure 9: **Machine versus human translations dev.** Left: Per-model rank correlation between language difficulty between MT and human translations, and min, max and average difference between the two conditions. Right: Difference between EM computed on human- and machine-translated data (human score - machine score), per language.

In Figure 9a we show the rank correlations of the difficulties of the various languages per model, as well as the min, max and average drop from human to machine translations. We see the that, at the model level, using machine translations rather than human translations results in a systematic undervaluation of the model scores: there is not a single model for which the 'drop' from human to machine translations is negative on average. In part, this is may be a result of the previously observed lack of knowledge transfer effect. That the drop is not substantially lower for models with better transfer, however, suggests that the more impactful factor is the quality of the machine translations, that may at times result in unanswerable questions.

In terms of model rankings, the difference between machine and human translations is minor: the model rankings between the two conditions have a rank correlation of 0.97 on the dev split, with only three local swaps (2&3 and 5&6 and 8&9) of models that did not have statistically different scores to begin with. This suggests that to compare models, using machine translation can be an acceptable alternative to human translations, as the mis-estimation is systematic across models.

Considering the effect across languages, we observe that even though the average drop is positive, for virtually all models there are at least some languages for which performance increases when MT is used, in some cases with even more than 10 points. For a handful of languages – specifically Russian, Swedish and Urdu – this is also true across models (see Figure 9b). While the overall rank correlation is high for language difficulty (0.88), it thus still urges caution in using machine translated data for language improvement prioritisation.

## F  Limitations

In this last section, we discuss various limitations of our work.

**Local relevance**  In our sourcing protocol, we explicitly sought to create questions locally relevant to the respective languages. It is important to notice, however, that some languages, such as English, Spanish, Portuguese, Chinese, French and to a lesser extent German and Dutch cover a wide variety of cultures. We did not separately control for that and the data for those languages thus likely comprises a mix of different locales.

**Data quality**  Building a bias-free evaluation datasets with few mistakes is not an easy feat. Even though we implemented several rounds of quality checks in our data collection pipeline, when looking at outputs we still incidentally found mistakes in the data or answers. We fixed some of these mistakes as we encountered them, but it is quite likely that more such mistakes occur in the dataset. It is also important to point out that we are less likely to spot such issues for languages that we do not understand at all, potentially creating a bias towards the set of languages for which we have a rudimentary understanding. Overall, however, we believe that the pipeline we designed assures a dataset of high quality. Of course, we welcome reports of mistakes spotted by others in the data.

**Evaluation**  Because MultiLoKo is a generative benchmark, computing scores requires comparisons of a generated answer with a set of gold answers. A first obstacle to this method of evaluation is that it is hard to create an exhaustive list of correct short-form answers. This is especially true when the correct answer is not a number, date, title or something else that can be expressed only in a few ways. In addition to that, it is hard to incentivise LLMs to produce concise answers. Even when instructed to answer with only a number / date / name / title, they may respond with a full sentence, add a reasoning trail to their answer, or add words beyond the minimal answer in a different fashion. We addressed such issues that were systematic in post-processing (see Appendix G), but it is hard to a priori catch all the ways that LLMs may deviate from the requested protocols. In some cases, we found additional post-processing steps that increased the scores of some models only later in the process, because scores for particular languages looked suspiciously low. For instance, we had not initially realised that our punctuation stripper did not strip punctuation in Urdu, which specifically influenced GPT4-o and Gemini. We considered several other metrics as well as judges, but eventually found that EM provided the clearest and least biased signal. It remains, however, a challenge to evaluate chatty LLMs completely independently from their ability to follow instructions.

**Wikipedia as information source**  MultiLoKo, as several other both multilingual as well as monolingual benchmarks, uses Wikipedia as main source of information. This has the advantage that Wikipedia has a large coverage across many different languages and the information is considered to be of high quality. It also facilitates comparable sourcing across languages. Of course, it also poses limitations. For one, it still provides a bias to the specific topics that can be included, that are usually primarily knowledge based. In fact, MultiLoKo is indeed a knowledge benchmark; it does not consider other types of skills. Secondly, and perhaps more importantly, Wikipedia is a a corpus frequently used in the training data of LLMs. The fact that MultiLoKo is a challenging benchmark even given that (multilingual) wikipedia is likely included in the training data of most of the LLMs evaluated suggests that this is not a large issue at the moment. However, it is very possible that MultiLoKo can be 'hacked' relatively easily simply by strongly oversampling multilingual wikipedia data.

# G  Instruction following

To facilitate evaluation, we instruct models to answer question with only a number/place/etc. Overall, we found that base models (with a five-shot template) are much better at abiding by this instruction than chat models, which exhibit a number of pathologies. While some of those can be caught with appropriate post-processing (see Appendix H, this is not the case for all issues. Below, we provide a summary of the main instruction-following issues we encountered with chat models.

**False refusals**  Sometimes chat models refuse to provide an answer when the question is falsely perceived to be inappropriate (e.g. when the question asks about someone aged younger than 18).

**Producing full sentences**  Another issue we observed is that chat models would provide a full sentence answer, rather than a single word or phrase (e.g. *Which year was Francisco Franco born? Produce a year only. – Francisco Franco was born in 1936*). Such full-sentence answers make exact match rating impossible. The effect is not consistent across languages and happens only for some of the examples, without any discernable pattern, and therefore difficult to address completely with post-processing.[7]

**Spurious addition of "answer is"**  Likely due to overtraining on MMLU style tasks, Models such as OpenAI's GPT4 and Gemini 2.0 preface the vast majority of the answers in English with "answer is" or "X answer is X" where X is the desired correct response. This is remarkable, because it is essentially a repetition of the end of the prompt. However, it is easy to fix in post-processing.

**Japanese specific issues**  In Japanese, in general it is not polite to answer with incomplete sentences. As such chat models often append the copula verb "desu" to the answer, making exact match unsuccessful. We are able to fix this in postprocessing.

**Claude 3.5 Sonnet issues**  We were unable to make Claude 3.5 Sonnet follow the instructions to produce just an answer in English. It seemed to engage in a long chain-of-thought reasoning style response which we were unable to reliably parse. This issue only manifests in English and only with Claude. For this reason, we exclude Claude 3.5 Sonnet from our knowledge transfer results, as it would make the average lack of knowledge transfer from non-English languages to English more severe than they are.

# H  Post-processing details

We perform the following post-processing for both the reference answers and the answers produced by the model:

- Remove leading and trailing whitespaces.
- Remove punctuation.
- Lowercase everything.

We perform the following additional post-processing for pretrained models:

- Remove leading "Answer:" or "A:" or the non-English equivalent from the output.
- Remove everything after the first newline.

We perform the following additional post-processing for postrained models:

- Remove leading "answer is:"
- Detect the pattern "X answer is X", where X is the desired answer, and strip the unnecessary part in the middle.
- Remove training "desu" in Japanese.

---

[7]Using a judge-LLM may to some extent address this problem, but at the expense of other issues (e.g. Thakur et al., 2024).

# I Annotation instructions

Our annotation pipeline contains five stages: 1) locality rating, 2) question generation 3) question review, 4) question answering, and 5) translation. Below, we provide the annotation instructions for each of these stages.

## I.1 Locality rating

To narrow-down the initial selection of paragraphs – sampled from the top-rated Wikipedia pages of the respective locales – the first step in our annotation pipeline is *locality rating*. Given a paragraph, we ask annotators to rate whether the paragraph is locally relevant to the particular locale, on a likertscale from 1 to 5, where 1 refers to extremely local and relatively obscure topics very specifically related to the specific language or locale and with little international recognition and 5 to globally well-known topics. We also ask annotators to disregard pages about inappropriate or politically sensitive topics. The rubric for locality annotation can be found in Table 4. We disregard everything with a locality rating of 3 or lower.

| | Description | Example |
|---|---|---|
| 1. | **Extremely local and relatively obscure.** Content that is of interest only to a small, localized group, such as a specific town, region, or community. These topics are typically obscure and not widely known beyond their immediate area. | Local radio stations, small town historical events, regional businesses, or niche local cultural practices. |
| 2. | **Regional interest.** Topics that have some relevance beyond a specific locality but are still primarily of interest within a particular region or country. | State or provincial politicians, regional cuisine, local sports teams, or medium-sized companies with regional influence. |
| 3. | **National Significance.** Content that is widely recognized within a single country, but relatively unknown internationally. | National politicians (not internationally known), popular national media figures, major corporations within a country, or significant national historical events. |
| 4. | **International recognition.** Topics that are recognized and have relevance in multiple countries but may not be universally known across the globe. These topics often have international influence and are likely to be covered in international media, though their impact may vary by region. | International brands which may be recognized in more than one country, celebrities with some international reach, significant cultural movements, or political conflicts with some awareness on the international stage. |
| 5. | **Global prominence.** Content that is widely recognized and relevant across a large number of countries around the world. These topics have a global impact or appeal and are likely to be well-represented in media across diverse cultures and regions. | Globally famous celebrities (e.g., Cristiano Ronaldo), multinational corporations (e.g., Apple), major world events, or universally recognized cultural icons. |

Table 4: **Rubric for locality rating task.** In the locality rating task, we ask the annotators to rate paragraphs with respect to how locally relevant the topic is to the locale.

## I.2 Question generation

The second and main annotation step in our pipeline is the step in which we ask annotators to generate questions about sampled paragraphs. We ask annotators to generate a challenging question with a short answer. The answer should be easy to evaluate with string-matching metrics, the questions should not be open-ended or have many possible correct answers, be ambiguous or subjective, and the expected short answer should be concise. To ensure difficulty, we ask that answering the question

requires combining information from different parts in the accompanying text; It should not be answerable by mere regurgitation of a single sentence. We furthermore ask that the question is formulated such that its answer will not change over time (e.g. not 'How many medals has Sifan Hassan won', but 'How many medals has Sifan Hassan won between 2018 and 2022 (including)'), and that the question is answerable also without the article (e.g. not 'How many tv shows did the person in this article produce?'). To facilitate validation checks in the next round, we also ask that the question authors write a longer answer to explain how they arrived at the short answer. We also ask the question authors to annotate what is the type of the correct answer (e.g. number, name, date, etc) In the pilot, we observed that – for some languages – the vast majority of questions were questions that required some form of numerical reasoning. Because the intention of the benchmark is to address knowledge more than reasoning, we afterwards restricted the number of numerical questions to 10%. Similarly, we asked question authors to avoid yes/no questions.

## I.3    Question review

In the first round of question review, we asked annotators from a different provider to judge whether the questions abide by the rules provided to the question authors. All question reviewers are native speakers. Specifically, we ask them to check if:

- The question pertains to a locally relevant topic

- The question is clear and undestandable, and not subjective

- The question has a clear and concise answer

- If there are multiple possible variations of the answer possible (e.g. 'Dick Schoof' / 'Minister Dick Schoof' / 'Prime Minister Dick Schoof' / etc), all versions of the answer are provided.

- The question and answer are in the correct language

- The question is understandable without the article

- That the answer to the question will not likely change in the near future

When a question can be fixed with a minor change (e.g. add a time indication to make sure an answer will not change in the near future, or add an extra answer version), we ask the question reviewers to implement this fix and describe it. In the pilot round, we use the annotator feedback to finetune our annotation protocol and provide feedback to the question-authors. During the rest of the data collection, we simply disregard questions that are not useable as is or can be corrected with minor changes.

## I.4    Validation through question answering

In the last stage of our question generation pipeline, we have additional annotators answer the sourced and reviewed question. The goal of this validation task is to confirm that the questions are answerable, correct, non-ambiguous when read by individuals other than the original question author, and that all possible versions of the answers are included. For each question, we ask two additional annotators to first answer the question, using the snippets the questions were sourced from for context. After they have answered the question, they are shown the list of reference answers written by the original author of the question as well as the rational they provided, and we ask them to reflect upon the answer they gave themselves. If their answer did not match any answer in the original reference list, we ask them to either add their answer to the list if it is semantically equivalent to their own answer or indicate which answer they believe to be correct, their own or the original answer. We disregard all questions where at least one annotator disagrees with the original question author.

