# OpenReview forum: "MultiLoKo: a multilingual local knowledge benchmark for LLMs spanning 31 languages"
_NeurIPS.cc/2025/Datasets_and_Benchmarks_Track — Submitted to NeurIPS 2025 Datasets and Benchmarks Track_

### Official Review · Reviewer_fEGh · 2025-07-01

**Rating:** 4
**Confidence:** 3

**Summary:**

The paper introduces MultiLoKo, a multilingual benchmark that evaluates large language models (LLMs) in 31 languages with questions of topics specific to the language. In addition to locally-sourced questions, MultiLoKo includes translation partitions that contain questions translated from other languages to English and vice versa. The authors evaluate 11 LLMs on this benchmark, reporting overall accuracy as well as the performance gap across different languages. They also compare scores on the locally-sourced and translation partitions to investigate mother tongue effects and locality effects.

**Dataset Code Accessibility:**

Yes

**Dataset Code Comments:**

It can be open-accessed on HuggingFace with standard formats.

**Ethical Considerations:**

No, there are no or only very minor ethics concerns

**Final Justification:**

The authors’ rebuttal addresses some of my concerns. However, they do not add an analysis on the failure case where correctly answered the question in one language but wrong in another. Therefore, I maintain my rating.

**Limitations Weaknesses:**

1. While the paper highlights cross-lingual inconsistencies in Section 5.2, it does not analyse the reason why models diverge on semantically identical prompts in different languages. A qualitative analysis of the failure cases would enrich the study.
2.	The paper does not provide sufficient examples of the questions in the benchmark. Including a small appendix or a table with representative questions would give readers a clearer sense of the benchmark’s challenges.
3.	Section 3.3 and Table 2 do not appear to cover every metric used in later analyses. For example, the consistency in Figure 3 and the rank correlation in Figure 4 are not included. Expanding the metric table to include all reported measures could make it easier for readers to follow the experimental results.

**Strengths Contributions:**

1. The paper clearly details the dataset collection process, incorporating multiple rounds of review to enhance the quality of both questions and answers.
2.	The authors conduct extensive experiments to evaluate the multilingual capabilities of LLMs. The investigation into mother tongue and locality effects provides valuable insights for the future multilingual evaluation.
3.	The presentation of the experimental results is clear. The figures effectively convey key insights to the reader.

---

> ### Author Rebuttal · Authors · 2025-07-30
>
> We thank the reviewer for their comments:
>
> - **Inconsistency between prompts**: Sadly this is a known issue that plagues LLMs. We don’t know of any consistent solution to this problem.
> - **Dataset examples**: The dataset is released publicly and is provided as part of the submission. Within the relatively limited space, we decided to prioritise results and analysis over a repetition of examples, but we encourage the reviewer to have a look at the examples on the github repository!
> - **Metrics** True, we cover only the main metrics that are explained in the text at that point, and not all the additional analyses. We could add the other ones as well, but would be a bit worried they are not understandable at that point. We would happily include them though if the reviewer doesn’t share that concern, do they think that it would be self-explanatory enough to add without also adding further explanations in section 3.3? Thanks!

---

> > ### Comment · Reviewer_fEGh · 2025-08-05
> >
> > Thank you for your rebuttal. While some of my concerns have been addressed, the analysis of the failure case remains insufficient. Therefore, I will maintain my original rating.

---

### Official Review · Reviewer_KHCw · 2025-07-02

**Rating:** 4
**Confidence:** 4

**Summary:**

- This work introduces a new benchmark called MultiLoKo, designed to test how well AI language models understand "local knowledge" across 31 different languages. They tested 11 major AI models and found that even the top-performing model only scored around 34% on average, with a huge performance gap between languages, indicating that models still have a long way to go to become truly multilingual.

- The core contribution is the MultiLoKo benchmark dataset itself. It's a high-quality, wide-coverage tool designed to address the "English-centric bias" of existing evaluation methods.

**Dataset Code Accessibility:**

Yes

**Ethical Considerations:**

No, there are no or only very minor ethics concerns

**Final Justification:**

I have maintained my score of 'Borderline Accept' after reviewing the author rebuttal. While the other three reviewers have rated it as 'Accept', I believe the paper possesses fundamental limitations that are not addressable during the rebuttal/revision phase. My decision is based on the following points:
- The concept of "region" is weak for global languages. The authors acknowledge this in their rebuttal ("For some languages, there is absolutely a case for multiple cultural inclusion"). Their response is that their benchmark is a "huge leap forward," an argument about its relative contribution, not a rebuttal of weakness. This is a critical design flaw. For major languages like Spanish, French, and Chinese, the benchmark lumps together diverse cultural contexts under a single language label, directly contradicting the paper's core premise: assessing fine-grained "local" knowledge. This isn't a problem that can be resolved with a single paragraph in the appendix; it requires rescoping and recollecting substantial amounts of data.
- The defense of the "Exact Match (EM)" metric is a standard but inadequate evasion. The authors claim they tried other metrics but found them "less informative" and included them in the appendix to improve "clarity." This is a common tactic. The fact remains that the entire paper's main results, figures, and conclusions rely on a fragile metric that the authors implicitly acknowledge have problems (e.g., having to exclude the Claude 3.5 Sonnet). Moving a failed analysis to the appendix does nothing to strengthen the main paper's argument.

**Limitations Weaknesses:**

- The greatest contribution of this study is its emphasis on “local relevance”, however, this core concept becomes blurred in its definition and operationalization when applied to global languages ​​that themselves span multiple cultures (such as Spanish, French, Chinese, etc.). The paper frankly admits this in Appendix F: “We did not control for different regional cultures within these (global) languages ​​separately, so the data for these languages ​​may contain mixed content from different regions”. This limitation weakens the “local” advantage claimed by the benchmark on these major languages. For example, local knowledge of “Chinese” can refer to knowledge from multiple different cultural regions such as mainland China, Taiwan, Hong Kong or Singapore. Lumping them together is contrary to the original intention of the precise cultural positioning that the benchmark aims to achieve.

- All questions in MultiLoKo come from Wikipedia. Although Wikipedia is a high-quality source of information, this singleness limits the type of "local knowledge" examined, biasing it towards well-documented, online-verifiable encyclopedic facts. This may overlook more subtle cultural common sense, social norms, or oral traditions that are not included in Wikipedia but are equally important. More importantly, the authors also acknowledge in the appendix that because Wikipedia is a common training data for LLM, model developers may "attack" this benchmark by specifically strengthening the training of multilingual Wikipedia, thus challenging its long-term effectiveness

- The study used "exact match" (EM) as the main evaluation metric, which is objective but may be too strict. An answer that is semantically correct but has slightly different wording will be judged as wrong. This is particularly problematic when evaluating generative dialogue models because it may confuse the model's "accuracy in following instructions" with its "degree of knowledge mastery." The paper mentioned that because Claude 3.5 Sonnet's answer was too "chatty" to be effectively parsed, it eventually had to be excluded from the partial transfer analysis, which just exposed the fragility of the EM evaluation method.

- The paper makes a strong case for different languages ​​being difficult for the model to learn, with graphs like Figure 8, but the analysis of the underlying reasons for these differences in difficulty is mostly hypothetical. What linguistic features (e.g., morphological complexity, syntactic structure) or data-level factors (e.g., scarcity of resources in the pre-training corpus) cause these differences?

-  If there is information loss, semantic deviation, or unnatural expression in the manual translation from the local language to English, then the low score on the English question may not be entirely due to the "poor knowledge transfer" of the model, but partly due to the "decreased input quality". Although the differences between "human translation" and "machine translation" are analyzed, there is no in-depth exploration of how the possible imperfections of "human translation" itself affect the interpretation of MTE indicators.

**Strengths Contributions:**

- The most important contribution of this study is the introduction of a new evaluation benchmark, MultiLoKo, which is innovative in that it fundamentally challenges and solves the inherent flaws of existing evaluation methods. Unlike popular benchmarks that rely on "English translation" content (such as MGSM), MultiLoKo uses 31 languages.

- MultiLoKo introduces new evaluation metrics such as "Mother Tongue Effect" (MTE) and "Locality Effect" (LE) through the combination of "local sourcing" and "bidirectional translation" (non-English - English). These metrics are important innovations because they can quantify deep problems that traditional accuracy metrics cannot capture.

- MultiLoKo conducted extensive testing on 11 state-of-the-art basic and dialogue models. All tested models performed poorly on the benchmark, with even the best-performing Gemini 2.0 Flash having an average accuracy of only 34.4%. More importantly, the "Gap" metric introduced in the paper - the difference in scores between the best and worst performing languages ​​- clearly exposed the serious imbalance in model performance between languages.

- The construction of the dataset has gone through multiple quality control steps, including "locality rating, multiple rounds of review, and verification answers by new annotators", to ensure the reliability of the data.

---

> ### Author Rebuttal · Authors · 2025-07-30
>
> We thank the reviewer for their comments and suggestions!
>
> - **Locality challenges**:  It is absolutely true that for some languages multiple cultures may be encompassed in the one language (e.g. French, Spanish). We do like to point out though, that locality always comes on a scale: even in different villages in the same country people may ask questions about different topics. To conclude, we believe that the benchmark is a large step up in terms of locality. It focuses specifically on content that is not globally relevant or English centric and is generated separately for each language by natives. We hope that in the future different researchers will develop more benchmarks with locally relevant knowledge with even more finegrained stratifications. (*NB*: for Chinese, the comment is not entirely correct. Wikipedia is not available in mainland China, and we have dedicated Cantonese evaluation for Hong Kong, Macau and (to lesser extent) Singapore and Malaysia.
> - **EM**: You are correct that EM has limitations, which is unfortunately an unsolved problem in LLM evaluation. We did implement several additional metrics to test the impact of this (specifically: chef, BLEU, contains, edit distance, edit similarity,  and even performed LLM-as-a-judge evaluations), and found the results not to be more informative than EM, so we decided for  simplicity/clarity to only report EM in the main paper. We will include some of these explorations in the appendix of the paper.
> - **Difficulty of languages**: we indeed do not consider studying this in the scope of this benchmark release, but do think the dataset could be beneficial to investigate these issues. If the reviewer is interested, there are several interesting studies on this topic, such as 2023.emnlp-main.658 and  2411.14198v1  (arxiv)
> - **MTE and translation**: This is true, even including the several conditions we did, it is very hard to disentangle everything. By including both human- and machine translations we hope to provide a dataset that can help researchers to further investigate this in the future, though we do believe it will remain an unsolved problem for quite some time. Personally, we do not think that effects this large would stem only from translation issues, but empirically establishing that would likely be extremely expensive. The fact that the human translations consistently have better scores, though, also points to that. We would like to note also that we did do quite some quality control on the translations. We used human translators from an internal provider that includes rigorous QA.

---

> > ### Comment · Reviewer_KHCw · 2025-08-04
> > **Response to rebuttal**
> >
> > Thank you for your response. I have read the rebuttal and the comments from the other reviewers. It has clarified several points.

---

### Official Review · Reviewer_bTHH · 2025-07-03

**Rating:** 5
**Confidence:** 4

**Summary:**

Paper introduces MultiLoKo, a new multilingual benchmark for evaluating large language models across 31 languages designed to address critical gaps in current multilingual LLM evaluation. It provides both a valuable evaluation tool and insights into how data sourcing decisions affect our understanding of model capabilities across languages that can help the research community better assess true multilingual competence of LLMs and identify areas for improvement in LLM training.
Paper main contribution is the dataset, prompts, evaluation code and leaderboard infrastructure that are all publicly available, allowing systematic analysis of language transfer and translation quality effects, that was indeed conducted in work very thoroughly with statistical rigour and adherence to best practices.
Dataset contains 500 human-written questions per language (15,500 in total), each question is sourced from a Wikipedia paragraph that is judged to be locally relevant, four parallel translation sets: human and machine translations from the 30 non-English languages into English and vice-versa and two splits per partition dev and hidden test. The test split uses lower-traffic Wikipedia pages to make it out-of-distribution.
Finally, the paper contains thorough study across 11 LLMs (4 base, 7 chat) that reveals current multilingual weaknesses and highlights pitfalls of relying solely on translated evaluation data. It demonstrates gaps in current SoTA and exposes poor overall performance and suboptimal knowledge transfer between languages as well as studies impact of data sourcing drawing multiple conclusions from differences between using locally-sourced and english-translated. It is discussing in depth results with regard to EM, Gap, MTE, LE and its consequences for multilingual benchmarking.

**Additional Feedback:**

Seems like untapped potential to include in research also French / Spanish and perform deeper analysis of knowledge transfer between more languages in graph like manner
Wonder if there is any way to include comparison of few shot vs classic approach on both chat and instruct models, ideally with different prompt versions to use and measure it's influence
Black & white print is hard to read - charts especially, and at times they would use slight redesign
Would recommend looking into work from Cohere esp. Aya

**Dataset Code Accessibility:**

Yes

**Dataset Code Comments:**

Requirements I think were missing jinja2 (and optionally pinning specific versions)
Also would be great to use tqdm or output text status that processing start to inform user creating examples when running python examples/create_examples.py

**Ethical Comments:**

No, used data from Wikipedia that is open and described annotation process involving humans that was following best practices

**Ethical Considerations:**

No, there are no or only very minor ethics concerns

**Limitations Weaknesses:**

Relying primarily on Wikipedia as the information source introduces potential biases and limits the types of knowledge tested. The authors acknowledge this limitation but it remains a constraint on the benchmark's scope.
Paper has limited task scope and focuses exclusively on factual knowledge questions. While this is clearly stated, the benchmark doesn't capture other important aspects of multilingual capability like reasoning, or cultural understanding beyond factual knowledge.
Language coverage could be greater as some major language families and regions remain underrepresented. Additionally, for languages covering multiple cultures (English, Spanish, etc.), the local relevance may be inconsistent. Also there is no explicit balancing across regional variants  - future expansion could stratify by locale to uncover intra-language biases.
Quality of human translation at times is worse than machine translation. Possibly multiple professional human translators could be used. It is good practice not to anchor annotators with displaying machine translation (although more expensive and hard to avoid at times esp. using Google translate). Also using solely Google translate for machine translation has non zero risk of introducing bias towards Gemini models - seems like an interesting thing to test by translating with other leading LLM providers.

**Strengths Contributions:**

Overall quality - paper is very clearly written, insightful and thorough
Addressing a critical gap in multilingual evaluation - paper tackles a fundamental limitation in current multilingual benchmarks - the English-centric bias from using translated data. The focus on locally-relevant content for each language is a significant methodological advance that better reflects real-world multilingual performance.
Rigorous data collection methodology - five-step collection protocol is well-designed, including locality rating, question generation, review, validation, and translation phases. The quality control measures with multiple rounds of annotation and validation help ensure dataset reliability.
Comprehensive experimental design - benchmark includes multiple partitions (locally-sourced, human-translated, machine-translated) enabling systematic comparison of different evaluation approaches. The analysis of 11 models provides substantial empirical evidence.
Excellent choice of evaluation metrics and practical relevance - directly useful for researchers needing language-parity diagnostics; findings can inform current evaluation practice, provides valuable insights into knowledge transfer and the impact of data sourcing choices on model evaluation as well as insights like the finding that using translated vs. locally-sourced data can change language difficulty rankings by up to 20+ points has important implications for multilingual model development and evaluation practices.
Transparency and reproducibility - authors provide detailed annotation instructions, release good quality code and data, and include comprehensive experimental details supporting reproducibility, even including experiment hardware setup and runtime details

---

> ### Author Rebuttal · Authors · 2025-07-30
>
> We thank the reviewer for their elaborate list of strengths and their constructive comments. Below we respond to the comments and questions of the reviewer.
>
> - **Wikipedia**: It is true that using only Wikipedia as information source limits the type of knowledge tested. Yet, we do think that the advantages of comparability and standardisation across languages and the fact that Wikipedia is a very large repository of information in this case outweigh the benefits. Furthermore, the fact that the scores are so low suggests that this is a type of knowledge relevant to assess. Of course, we welcome other benchmarks that focus on knowledge beyond what is found in Wikipedia!
> - **Scope (task, number of languages, further stratification beyond languages)**: While it is undeniably true that one can always add more types of tasks, more data and more languages, we contest that not doing so really constitutes a constraint on the benchmarks scope. Evaluating multilingual reasoning would be valuable as well, as would be many other things in the multilingual space (conversationality, speech ability, math, etc, etc). Yet, it is in the nature of benchmarks that they focus on part of the problem space, which in our case is (local) knowledge.
> - **Language transfer**: Given that we already had to move a lot of our analysis to the appendix, we decided not to add even more (though we briefly talk about it in Figure 3). That said: we do agree that it is a very interesting topic and that our benchmark is very suitable to study it. We see it as a natural future work extension and hope that it will be picked up in the future (either by us or other researchers).
> - **Translation quality**: It is absolutely true that whatever type of translation has biases. In part because of this reason, the main partition of our benchmark does contain only data that is collected from scratch in the respective language and the main metrics of the benchmark (average EM & gap) use only that data. We hope that the translated partitions, covering both human- ánd machine translated data, will help future researchers further research the impact of using translations and study the difference between machine- and human translated data when it comes to LLM evaluation.

---

> > ### Comment · Reviewer_bTHH · 2025-08-08
> > **Still accept**
> >
> > Thank you for response. I accept your rebuttal, and after reading it as well as comments from other reviewers (that shared my concerns) and your explanations, I decided not to lower my rating

---

### Official Review · Reviewer_NsJN · 2025-07-06

**Rating:** 4
**Confidence:** 5

**Summary:**

The work tries to address limitations in existing multilingual benchmarks by creating MultiLoKo a multilingual benchmark in 31 languages to evaluate LLMs. The benchmark consists of 500 questions per language sourced to be locally relevant and human-authored translations from 30 non-English languages to English and vice versa. The goal is to provide a better means to evaluate Multilinguality in LLMs and to provide data to study the effect of various design choices in multilingual evaluation.

**Additional Feedback:**

- In reference to Figure 5, more clarification would help to asses the conclusion made that the test set is indeed out of distribution with respect to the dev set. Given that the em score difference between the dev and test set is 3 and the gap between the best and worst languages is 37 and 34, respectively for the dev and test set.

- when calculating the average LE score, would be helpful to calculate the mean square or root mean square or even using absolute values to ensure that both positive and negative LE scores are well represented in the data. Otherwise in the case when a model has an LE of -3 and another model has an LE of +3 the average LE score is 0 which isn’t very meaningful or a good indicator of the actual results.
- It is clear that base models were better than chat-models, this might be due to the fact that base models were evaluated on a five-shot evaluation, unlike chat models evaluated on a zero shot setting. Could poor promts and false refusal be the reason as mentioned in the paper? Would it help to try prompting variants and figure out the best prompts or even average out the scores of several semantically-identical prompts for efficiency purposes i.e., example, for each prompt or question, generate 5 different paraphrasing of the same prompt and evaluate the model on the five prompts.

**Dataset Code Accessibility:**

Yes

**Dataset Code Comments:**

The dataset is readily available and accessible via huggingface hub here: https://huggingface.co/datasets/facebook/multiloko

**Ethical Considerations:**

No, there are no or only very minor ethics concerns

**Limitations Weaknesses:**

Some of the limitations include:

- The EM metric used for evaluation is rigid and could penalize semantically correct answers especially with chat-models which have been fine-tuned on specific answer prompts.
- Annotators demographics is not explicit, the authors mention that people who generated the questions were native speakers. More details about the annotators would be helpful.
- MultiLoKo uses 250 most visited wikipedia pages in its dev set and 250 least visited pages for testing, understanding the difference between the most visited and the least visited pages content is crucial. Given that commonly visited pages are usually about trending topics like global news and celebrities which use fairly easy language and can be understood by most people without any prior knowledge or effort, least visited pages are usually about very niche science or historical topics that use very hard or uncommon jargon, and they are usually hard to understand by non-experts. With this setup, it’s very hard to tell if this benchmark accurately assess the model’s ability to generalize on unseen data.

**Strengths Contributions:**

- One of the key contributions is the idea of the Mother Tongue Effect, which suggests that the quality and cultural relevance of evaluation data can outweigh stronger English-language performance in assessing multilingual models. This insight highlights the importance of designing benchmarks grounded in authentic, culturally situated language use.

- The paper is generally well-written, logically organized, and easy to follow. Figures and tables are informative and support the key findings effectively. Overall, the work is a significant and thoughtful contribution to the multilingual NLP evaluation and creation of benchmarks.

- The authors also clearly describe existing studies and how their study deffer from existing work, the figures and tables are clear and descriptive.

---

> ### Author Rebuttal · Authors · 2025-07-30
>
> We thank the reviewer for their comments and are happy with their positive notes about our work :)
>
> Below a respond to their comments/questions:
> - **EM metric**: You are correct that EM has limitations, which is unfortunately an unsolved problem in LLM evaluation. We did implement several additional metrics to test the impact of this (specifically: chef, BLEU, contains, edit distance, edit similarity,  and even performed LLM-as-a-judge evaluations), and found the results not to be more informative than EM, so we decided for  simplicity/clarity to only report EM in the main paper. We will include some of these explorations in the appendix of the paper.
> - **Demographic stats of the annotators**: good suggestion! We reached out to the vendor and will report these in the final version of the paper
> - **Visited pages**: We believe that the reviewer has slightly misunderstood the split procedure. We first selected the top 6000 articles for each language, from which we selected articles that were locally relevant until we had enough, starting with the most popular articles and moving down. From those selected articles (which are usually not near the bottom of the top 6000), we then asked annotators to generate locally relevant questions. The questions created from the “bottom 250” articles are thus oftentimes still questions pertaining quite common topics that people engaged in that culture would be familiar with. In addition to that, we would like to point out that the purpose of the OOD test set is not to assess models' ability to generalise to unseen knowledge/pages. In fact, we do not think that is possible. Instead, the OOD test set is meant to assess whether there is generalisation to tail knowledge, which is -- indeed -- more niche (and also to understand if modelers are not overfitting on the dataset). Thus, even if the knowledge in the less frequently visited 250 is more obscure or more difficult, that does not defeat the purpose of the ood test set: we **want** that set to have distributionally different properties than the dev set.
> - **LE score calculation**: this is an interesting suggestion. We had initially thought about this, but given the fact that a negative and positive LE score have quite a different meaning, we instead opted for displaying the average LE per language, along with the standard error which provides a rough indication of the different values. This allows us to see that for some languages the locality effect is strongly negative (even when taking the average across models!) whereas for other languages the locality effect is very positive.
> - **Prompts for chat models**:  This is a reasonable point. Unfortunately, the instruction following-capabilities of post-trained models are not yet at a level that they can answer any question out of the box. For some languages that we understood ourselves, we did experiment with prompting to understand what type of prompts and patterns worked best for the models evaluated. Doing extensive prompt analysis for each language was unfortunately not feasible given the language expertise that we had available. However, we have released the prompts and the benchmark is open, and we do really hope that natives will help improve upon the prompts!

---

> > ### Comment · Reviewer_NsJN · 2025-08-05
> >
> > I thank the authors for the feedback, I have read the comments for the authors and since some of the concerns I raised will be addressed in the next version, I have decided to maintain my original rating.

---

### Decision · Program_Chairs · 2025-09-18

**Decision:**

Reject

**Comment:**

MultiLoKo introduces a rigorous and much-needed multilingual benchmark spanning 31 languages, with locally sourced questions, human and machine translations, and multiple partitions enabling analysis of mother-tongue and locality effects. The work is timely, addresses the English-centric bias of existing evaluation, and provides thorough experiments across 11 LLMs, showing large gaps in multilingual capabilities. While limitations exist (e.g., reliance on Wikipedia, challenges in defining “locality” for global languages, strictness of EM metric), the dataset, code, and leaderboard are openly released and reproducible, and the authors engaged constructively with reviewer concerns. Given its clear novelty, breadth, and community value, I recommend acceptance.

===== FINAL UPDATE FROM DB Track PCs ====

The final decision for this paper has been taken by the program chairs after consultation with the SACs. All Senior Area Chairs have ranked papers according to the feedback from the AC during the review process. We decided to leave the original meta-review to reflect the opinion of the AC in light of the initial discussions with reviewers and SAC.